# Polymer Pen Lithography-Fabricated DNA Arrays for Highly Sensitive and Selective Detection of Unamplified *Ganoderma Boninense* DNA

**DOI:** 10.3390/polym11030561

**Published:** 2019-03-25

**Authors:** Ekta Rani, Siti Akhtar Mohshim, Muhammad Zamharir Ahmad, Royston Goodacre, Shahrul Ainliah Alang Ahmad, Lu Shin Wong

**Affiliations:** 1Manchester Institute of Biotechnology and School of Chemistry, University of Manchester, 131 Princess Street, Manchester M1 7DN, UK; ades.ekta@gmail.com (E.R.); roy.goodacre@liverpool.ac.uk (R.G.); 2Department of Chemistry, Universiti Putra Malaysia, Serdang 43400, Selangor, Malaysia; ctakhtar@gmail.com; 3Biotechnology and Nanotechnology Research Centre, Malaysian Agricultural Research and Development Institute, Serdang 43400, Selangor, Malaysia; zamharir@mardi.gov.my; 4Institute of Advanced Technology, Universiti Putra Malaysia, UPM Serdang 43400, Selangor, Malaysia

**Keywords:** polymer pen lithography, sandwich assay, genomic DNA, visual and optical detection, *Ganoderma boninense*

## Abstract

There is an increasing demand for lithography methods to enable the fabrication of diagnostic devices for the biomedical and agri-food sectors. In this regard, scanning probe lithography methods have emerged as a possible approach for this purpose, as they are not only convenient, robust and accessible, but also enable the deposition of “soft” materials such as complex organic molecules and biomolecules. In this report, the use of polymer pen lithography for the fabrication of DNA oligonucleotide arrays is described, together with the application of the arrays for the sensitive and selective detection of *Ganoderma boninense*, a fungal pathogen of the oil palm. When used in a sandwich assay format with DNA-conjugated gold nanoparticles, this system is able to generate a visually observable result in the presence of the target DNA. This assay is able to detect as little as 30 ng of *Ganoderma*-derived DNA without any pre-amplification and without the need for specialist laboratory equipment or training.

## 1. Introduction

The demand for the fabrication of micro- and nanoscale features of biomolecules such as proteins and oligonucleotides for their applications in biosensing, diagnostics, and tissue engineering is increasingly widespread [1,2,3]. However the lithography of nanoscale features consisting of “soft” biomolecules remains a significant challenge. Photolithography and electron beam lithography are staples of conventional nanofabrication in the semiconductor industry, but they are difficult to adapt to biomolecules because of the harsh (“hard”) processing conditions used, which can include the use of short wavelength ultraviolet light or ultrahigh vacuum conditions. A common soft lithography approach that has been demonstrated for the patterning of biomolecules is microcontact printing (μCP), which uses an elastomeric stamp to print the molecules on to the surface [4,5,6]. μCP offers an easy and inexpensive route for stamping soft molecules over large (cm^2^) areas, under ambient conditions. However, feature fidelity is generally poor at the sub-μm scale. Furthermore, there is a lack of design flexibility, since the pattern is fixed by the design of the photolithographically fabricated master.

As an alternative, scanning probe microscopy-based methods such as dip pen lithography (DPN) offer the generation of biomolecular features with sub-100 nm resolution at or near ambient conditions [7,8]. DPN has also been further developed to use large arrays of probes to enable parallelised large-area fabrication [9]. However, these large “two-dimensional” DPN probe arrays are technically difficult to produce and delicate to use. To address these limitations, polymer pen lithography (PPL) was developed [10,11]. Instead of the cantilevered arrays of stiff probes used in DPN, PPL employs arrays of pyramidal probes consisting of silicone elastomer that are robust and inexpensive. PPL thus combines the advantages of µCP and DPN: the ability to pattern large areas with arbitrary patterns of soft materials, down to sub-100 nm resolution; simple probe array preparation and robustness of use. Indeed, PPL has previously been demonstrated for the deposition of both proteins [12] and oligonucleotides [13].

In terms of end applications, it has been proposed that “arrays” of oligonucleotides offer a method for the detection of specific DNA sequences, and thus the potential for the creation of medical diagnostic kits [14,15]. Patterning of microarrays of biomolecules is limited in application due to the requirement for relatively large sample volumes, prolonged incubation time, and a relatively poor limit of detection. Arrays of nanoscale features can offer better performance with detection down to ~10^6^ copies compared to microscale patterns (with features typically in the 10s of μm) [16] or non-patterned surfaces (i.e., whole surface covered) [17]. Apart from the device performance, such oligonucleotide arrays also offer the benefit of ease of use (without the need for specialist training or equipment), portability and tolerance to a wide range of environmental conditions. These aspects are particularly important when considering applications in the agricultural sector and in the developing world, which typically require diagnostics outside specialist laboratories (i.e., in the field).

As a specific example of an application in tropical agriculture, there is a need for convenient methods for the detection of the fungal pathogen *Ganoderma boninense*, which causes basal stem rot disease in oil palms [18,19,20]. Once established, this disease is the major cause of yield loss in this crop, and early detection would enable isolation of the infected palms [18]. However, early diagnosis is difficult because the disease is symptomless in early infection and *G. boninense* has various resting stages where there is apparently little change to the host plant [20]. Furthermore, *G. boninense* is a soil-borne organism, and its detection in soil samples would also enable the growers to manage the land by fallowing or sanitation before replanting [20].

Molecular methods to detect *G. boninense* based on antibodies [21,22,23] and electrochemical sensors [24,25,26,27,28] have been employed. Various non-molecular methods based on density and ultrasonic detection [29], spectral imaging [30,31,32], and backscattering measurements [33] have also been used in an attempt to detect *G. boninense*. However, these detection methods are either not species-specific, and/or possess practical disadvantages in regards to complexity, cost and the need for specialist training.

Herein, the use of PPL to produce oligonucleotide arrays is reported, which together with DNA-gold nanoparticle (DNA-AuNP) conjugates, provide a sensitive and selective detection of unamplified genomic DNA of *G. boninense* extracted from mycelial cultures of this organism. Using a sandwich assay format [34], in the presence of the target DNA sequence from the target organism, the complementary “capture” DNA-AuNPs are immobilised on to the “reporter” surface arrays (Figure 1). This immobilisation results in an alteration of the surface properties that can be visualised optically, and with high sensitivity and selectivity.

## 2. Experimental Section

### 2.1. Materials and Equipment

All chemical reagents were purchased from Sigma Aldrich, Merck or Thermo Fisher Scientific and used as supplied. All reagents were of analytical grade. 20 nm AuNP colloid (in 0.1 mg mL^−1^ sodium citrate) at a concentration of 1 optical density (OD) unit @ 520 nm were purchased from Kestrel Biosciences (Pathumthani, Thailand). Sterile deionised water was used for all the experiments. High-performance liquid chromatography (HPLC) purified single-stranded DNA (ssDNA) of the target sequence, “capture” (with a 5′ S-S C6 disulfide linker) and “reporter” (with a 3′ S-S C3 disulfide linker) were purchased from IDT and supplied in the lyophilised form. The target ssDNA sequence corresponds to a fragment of the *G. boninense* internal transcribed spacer 1 (ITS1) within the 18S ribosomal gene with the National Center for Biotechnology Information (NCBI) GenBank accession number EU701010. Sequences of the target, capture and reporter ssDNA are listed in Table 1. 

NEXTERION^®^ glass slides were purchased from Schott (Jena, Germany). Au film deposition was performed using BOC Edwards Auto 500 electron beam deposition system. The O_2_ plasma treatment was performed using Harrick Plasma PDC-32G plasma cleaner (Ithaca, NY, USA). Spin-coating was performed with a Laurell Technologies WS-650-23NPP spin-coater (North Wales, PA, USA). Polymer pen lithography and atomic force microscopy (AFM) were performed on a custom-built atomic force microscope with a 5-axis sample stage (Nanosurf AG, Liestal, Switzerland) equipped with an automated alignment algorithm reported previously [35,36].

The non-complementary genomic DNA was extracted using an Exgene Plant SV Mini Kit (GeneAll Biotechnology Co. Ltd., Seoul, Korea). Sonication of the extracted DNA was performed with Qsonica Q125 sonicator (Cole-Parmer, Vernon Hills, IL, USA). A NanoDrop ND-1000 (NanoDrop Technologies LLC., Wilmington, DE, USA) was used to measure the concentration of genomic DNA at 260 nm.

### 2.2. Preparation of Capture DNA-AuNP Conjugates

Capture DNA-AuNP conjugates were prepared as previously described [37] with a few modifications. Specifically, the capture probe ssDNA was dissolved in water to a concentration of 100 μM. A 65 µL aliquot of the capture probe DNA was mixed with 34 µL of sterile deionised water and 1 µL of 10 mM aqueous tris(2-carboxyethyl)phosphine hydrochloride (TCEP) solution and incubated at room temperature (RT: ~27 °C in Kuala Lumpur) for 1.5 h to reduce the disulfide bond in the capture DNA. The mixture was then added to 1 mL of AuNP colloid and incubated at RT for 16 h. 135 µL of sterile deionised water was added to the mixture. 13 μL of 5 M NaCl was added to this mixture and subjected to ultrasonic agitation for 10 s, then allowed to stand at RT for 1 h. This process of addition of NaCl, agitation and standing were performed five times in total, until a final concentration of 0.25 M NaCl was achieved. The final colloidal solution was allowed to stand at RT for 24 h and then centrifuged at 17,000× *g* for 30 min. The supernatant was decanted, and the pellet was resuspended in phosphate-buffered saline (PBS) diluted to half-strength (i.e., final composition 5.9 mM phosphate buffer, 68.5 nM NaCl, 1.35 mM KCl, pH 7.4) containing 0.01 % *v/v* Tween 20. This process of centrifugation, decanting and resuspension was repeated three times to remove the unreacted DNA. The final pellet of AuNP was re-suspended in 300 μL of PBS buffer (pH 7.4).

### 2.3. Preparation of Reporter DNA Arrays by PPL

#### 2.3.1. Preparation of Materials

10 nm Au films (with a 2 nm Ti adhesion layer) were deposited on to glass slides by electron beam deposition according to standard procedures at a rate of 0.03 nm s^−1^ under 5 × 10^−6^ Torr. The elastomeric PPL probe arrays were prepared according to the procedures previously reported [35]. For the purposes of this research, probe arrays of 13 × 13 cm^2^ were used with an interprobe pitch of 100 µm. Prior to lithography the probe arrays were treated with O_2_ plasma (500–600 mTorr at maximum RF power for 2 min).

The reporter ssDNA that was to be printed was prepared by reconstituting the lyophilised DNA disulfide with deionised water (to a concentration of 100 µM). 49 µL of this solution was then mixed with 1 µL of 20 mM aqueous solution of TCEP for 2 h to reduce the disulfide bond. Two formulations of “ink” solution containing this TCEP-treated DNA were then prepared and deposited on to the PPL arrays:

Ink **1**: The reporter ssDNA solution was diluted to a final concentration of 50 μM in a solution containing 90% DMF/10% water and 0.3 M MgCl_2_. The PDMS probe array was dipped for 10 s into this solution and dried under a stream of N_2_ gas; or spin-coated at 3500 RPM for 3 min [38].

Ink **2**: The reporter ssDNA solution was diluted to a final concentration of 50 μM in a trehalose-containing buffer (400 mM potassium phosphate buffer of pH 7.0, 0.5% *w/v* trehalose dihydrate, 0.1% *v/v* Tween 20, and 20 % *v/v* glycerol). 20 µL ink was dropped onto the PPL array and spin coated at 3500 RPM for 3 min [13]. 

#### 2.3.2. PPL Printing of Reporter DNA Arrays

The ink-coated probe array was attached to the middle of the probe holder, and the probe holder was mounted on to the AFM. The Au coated glass slide was mounted in the middle of the AFM sample stage and the alignment of the probe array with the slide was performed [35] with the “Angle Step” parameter set at 0.15° and the “Coarse Step” and “Fine Step” settings at 0.6 and 0.2 μm, respectively. After the alignment process was completed, the AFM atmospheric isolation chamber was engaged and set to a relative humidity of 65%, a temperature of 23 °C and a gas flow rate of 500 mL min^−1^. Once desired humidity was reached, the coated probes were moved to the desired position on the substrate and the lithography was performed for the chosen pattern with a dwell time of 1 s per dot feature. Two different patterns were printed whereby each probe produced; (i) a single grid of 25 × 25 dot features with an interfeature pitch of 3 µm; (ii) a 2 × 2 matrix of arrays (distance between two grids is 20 µm), with each array consisting of a 10 × 10 dot features with a pitch of 2 µm. Post-lithography, the gold substrates were immediately imaged by AFM and optical microscopy, before being washed with water and dried with an N_2_ stream.

The substrates used for subsequent hybridisation experiments were immersed in a solution of triethylene glycol mono-11-mercaptoundecyl ether (1 mM in ethanol) for 1 min and air dried. 

### 2.4. Extraction of Genomic DNA

Fruiting bodies of *Ganoderma* were collected from an oil palm plantation located in Lahad Datu, Malaysia. Isolation of the pure culture of *G. boninense* was carried out on potato dextrose agar plates using a previously published protocol [39]. The samples were incubated in the dark at 25–28 °C during growth of the culture. 

Extraction of the genomic DNA was performed using a previously reported protocol [40] with some modifications. In brief, approx. 100–500 mg of the fungal mycelia was homogenized using a disposable micro pestle with 400 µL DNA extraction buffer (200 mM Tris, 250 mM NaCl, 25 mM EDTA, 0.5% *w/v* SDS, pH 8.5) and the homogenate shaken at 65 °C for 10 min. 130 µL of 3 M sodium acetate (pH 5.2) was added, the lysate incubated at −20 °C for 10 min, then separated by centrifugation (16,100× *g* at 4 °C for 15 min). The clear supernatant was decanted, into which an equal volume of isopropanol was mixed and incubated at −20 °C for 10 min. This mixture was then centrifuged (3300× *g* at 4 °C for 20 min), the supernatant discarded and the pellet was washed with 100% and then 70% ethanol. The pellet was air dried for 10 min at 40 °C and resuspended in 100 µL of sterile deionised water. 

The non-complementary genomic DNA was extracted from soya beans using DNA extraction kit according to the kit manufacturer’s instructions. 

The extracted DNA was subjected to sonication on ice (25% of amplitude, 10 s pulse and 3 s rest) to shear the DNA into smaller fragments [41]. Concentration of genomic DNA was measured post-sonication by ultraviolet–visible (UV–Vis) spectrometry.

### 2.5. Hybridisation Experiment on PPL Arrays

For the hybridisation experiment, 100 μL of capture DNA-AuNP colloid (from Section 2.2) was centrifuged at 17,000× *g* for 10 min, the supernatant discarded and the pellet resuspended in 99 μL of hybridisation buffer consisting of 10 mM sodium phosphate buffer, 300 mM NaCl at pH 7.0. To this mixture, 1 μL of target ssDNA was added for the hybridisation experiment. 

For genomic DNA, the pellet of AuNP was resuspended in 97.5 μL of hybridisation buffer and 2.5 μL of genomic DNA was added for the hybridisation experiment. 

This solution was allowed to stand under the appropriate conditions (see Section 3.2 and Section 3.3) and was then dropped on to a PPL-printed reporter DNA array slide and incubated for the appropriate time. The slide was then washed with 1 M ammonium acetate and air dried. Post-hybridisation patterns were visualised using an optical microscope or visual examination. 

## 3. Results and Discussion

### 3.1. PPL Fabrication of Reporter DNA Probe Arrays

In order to produce the DNA arrays, PPL was employed to print the thiolated DNA sequences on to gold-coated slides, as this enabled the convenient and rapid generation of cm^2^ areas of patterns. For this purpose, two DNA “ink” formulations were tested, which were based on previous reports of DNA printing with either DPN [38] or PPL [13] (Ink formulation **1** and **2** respectively; see Section 2.3.1). 

Ink **1**, consisting of a DMF/aqueous formulation, was generally found not to wet the PPL probe arrays uniformly and thus did not result in the deposition of the DNA on to the gold surface, as evidenced by the lack of observable features by either AFM or optical microscopy (Appendix A). This same result was observed regardless of whether the ink was dip- or spin-coated on to the probe arrays. 

Next ink **2** was tested, which consisted of a buffer with trehalose added as a wetting agent. In contrast to ink **1**, spin coating of ink **2** gave a uniform spreading of the ink on to the probe arrays. Using this ink loading method, the subsequent PPL of a 2 × 2 grid matrix (each grid consisting 10 × 10 dots) was performed. AFM imaging post-lithography clearly displayed the desired patterning (Figure 2a,b), which could also be visualised by optical microscopy (Figure 2c). Under the lithography conditions used, the individual dots with a full width at half maximum of ~ 1.0 ± 0.2 µm were observed by AFM. In the negative control experiment, when the same lithography procedure was performed with the buffer mixture but with the DNA omitted, no features were observed by either AFM or light microscopy. The surface substrates were then washed with water and subsequent imaging showed that these features were no longer visible by optical microscopy. This suggests that features observed before washing are composed primarily of the trehalose and buffer salts along with reporter DNA, which upon washing removed these excess ink components, leaving the monolayer of thiolated DNA on the surface. Notably, the features were also no longer observable post-washing even by AFM, which was attributed to both a decrease in feature height and increased roughness of the substrates post-washing. 

Prior to the hybridisation experiments, the printed and washed substrates were immersed in a solution of triethylene glycol mono-11-mercaptoundecyl ether in order to passivate the unprinted areas and prevent non-specific adsorption of biomolecules. 

### 3.2. The Hybridisation of Patterned Substrates with Single-Stranded Target DNA

As an initial evaluation of the arrays, the detection of a short synthetic ssDNA fragment corresponding to a *G. boninense* gene was attempted. Adapting a previously reported procedure [38], two possible approaches were tested: (i) mixing of the target DNA sample with the capture DNA-AuNP conjugates and direct application on to the PPL-printed reporter DNA arrays; or (ii) incubating the target DNA and DNA-AuNP conjugates overnight before application on to the arrays. In both cases, 10 pmol of target ssDNA (10 μM in 1 μL of analyte sample made up to 100 μL of hybridisation solution added to the array) and arrays with continuous 25 × 25 dot grids were used. After a 3 h exposure to the hybridisation solution, the arrays were washed and examined by optical microscopy. 

It was found that faint features were observable only when the target DNA was pre-incubated overnight with the DNA-AuNPs prior to immobilisation on to the array slide (Figure 3a), while the procedure that did not involve this incubation step did not give any features. The poor visibility was thought to be due to insufficient binding of the target (and pre-hybridised reporter DNA-AuNPs) on to the reporter DNA arrays. Thus, in order to improve the visibility of the features, after the mixture was drop-casted on to the array a heating step was incorporated, either 30 °C for 3 h, 50 °C for 3 h or 50 °C for 1 h; followed by a cooling period of 10 min at RT prior to examination by optical microscopy. It was found that only the experiment involving the 30 °C heating gave clearer features that were organised into grid-like squares that were consistent with the PPL-printed pattern (Figure 3b). The other two heating regimes gave no visible features at all, suggesting that excessive heating prevented binding to the array.

Similar results could also be achieved with 5 pmol (5 μM in 1 μL of analyte sample) corresponding to approx. 3.0 × 10^12^ copies of the gene (Figure 3c). These features remained visible even after several washes, indicating that the attachment to the arrays was robust. Two negative control experiments were also performed, whereby only the capture DNA-Au NP conjugates or only the target DNA were added to the arrays. In both these cases, no features were observed (Figure 3d,e). 

Hybridisation of the complementary target DNA with arrays were also observable by the naked eye as a darker area on the array when it was held at a ~45° angle (Figure 3f), which was clearly differentiated from the unprinted area.

### 3.3. Detection, Sensitivity and Selectivity for Genomic DNA of G. Boninense

Having demonstrated the general procedure and generation of these arrayed nanomaterials, the detection of genomic DNA extracted from cell culture was then investigated. 

In the initial investigation a 400 ng (in a 2.5 μL sample) of genomic DNA was tested, corresponding to approx. 5.8 × 10^6^ copies of the *G. boninense* genome (see SI for details of calculations). However, in this case the mixture of the sample DNA and capture DNA-AuNPs was heated to 95 °C for 5 min (to denature the double-stranded genomic DNA and enable binding with the capture DNA) and allowed to cool for 1 h, prior to deposition on to the reporter array as noted above (see Section 3.3). Optical microscopy post-washing showed the presence of dot patterns with good contrast corresponding to the originally PPL-printed pattern (Figure 4a), which indicated successful capture of the genomic DNA. 

To further investigate the sensitivity of this method, a further series of genomic DNA amounts (160, 30 and 3 ng) were tested. Positive results were obtained for 160 ng (2.4 × 10^6^ copies) and 30 ng (4.4 × 10^5^ copies), but not for the lowest amount of genomic DNA (Figure 4b,c). However, it should be noted that the observed detection range is subject to a degree of uncertainty for at least two reasons. Firstly, each copy of the *G. boninense* genome includes many copies of the ribosomal 18S gene, of which the absolute number is poorly defined (typically 100s) [42]. Secondly, the extracted DNA samples are subjected to ultrasonic shearing in order to generate smaller fragments that are compatible with this assay. If the fragmentation occurs within the target sequence, these fragments are unlikely to undergo hybridisation with the capture and reporter probes. Nevertheless, this level of sensitivity is likely to be facilitated by ordered orientation [43] of the capture and reporter strands, which enable efficient hybridisation and immobilisation of the target strands. 

Finally, in order to study the selectivity of this method, a sample of genomic DNA that does not contain a complementary sequence to the capture and reporter sequences was tested using DNA samples extracted for soy beans. Here, even samples containing up to 225 ng of soya bean DNA gave a negative result, as evidenced by a lack of observable features on the array (Figure 4d).

## 4. Conclusions

In summary, a method has been developed for the label-free, sensitive and sequence selective detection of DNA sequences from biologically extracted samples, without the need for polymerase chain reaction (PCR) amplification of the sample DNA material. Crucially, hybridisation of complementary target/genomic DNA with DNA arrays and DNA-AuNPs can be observed with the naked eye. As an exemplar, positive results are obtained down to approx. 30 ng of genomic DNA from *G. boninense* mycelial tissue. This result is significant in comparison to other methods, which typically require the use of the PCR for the amplification of the DNA samples to produce sufficient copy numbers for detection [44,45].

This method relies on a “sandwich” assay where the target DNA is immobilised between DNA-AuNP conjugates and a DNA array on a surface. Key to the development of this method are the DNA arrays that were rapidly and conveniently fabricated using PPL. The patterned surface generated by this lithography method facilitated clear identification of a positive result by presenting a clear contrast between the printed areas where the AuNPs aggregated, and the unprinted areas. Although the fabrication of the DNA arrays involves the use of PPL that does require specialist equipment and training, once the arrays are produced these are stable, portable and easy to use in the field.

In principle, a further enhancement of the assay sensitivity could be obtained through the use of silver enhancement of the deposited AuNPs [34]. The current limitations of this assay are the need for some pre-treatment of the DNA and a heating step for dehybridisation. Future work will, therefore, be aimed at the development of chemical and isothermal methods for DNA processing, to enable its application directly from homogenised tissue samples. 

## Figures and Tables

**Figure 1 polymers-11-00561-f001:**
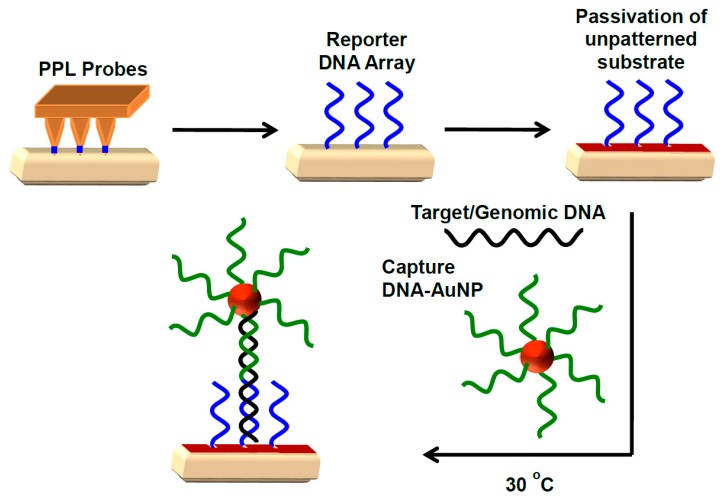
Schematic diagram illustrating the immobilisation of reporter DNA using polymer pen lithography (PPL) probes on a Au surface followed by the passivation of unpatterned substrate with triethylene glycol mono-11-mercaptoundecyl ether. In the presence of capture DNA-AuNPs, the target DNA forms a sandwich assay that can be detected optically.

**Figure 2 polymers-11-00561-f002:**
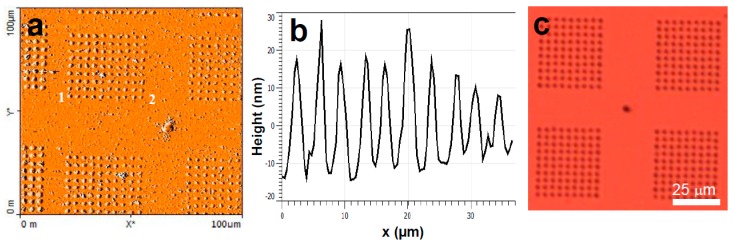
(**a**) Representative atomic force microscopy (AFM) topography image of a substrate patterned with reporter DNA dissolved in trehalose buffer, (**b**) representative corresponding cross-section profile along the line between points 1 and 2 marked in (**a**), and (**c**) representative optical microscopy image of a substrate patterned with reporter DNA dissolved in trehalose buffer.

**Figure 3 polymers-11-00561-f003:**
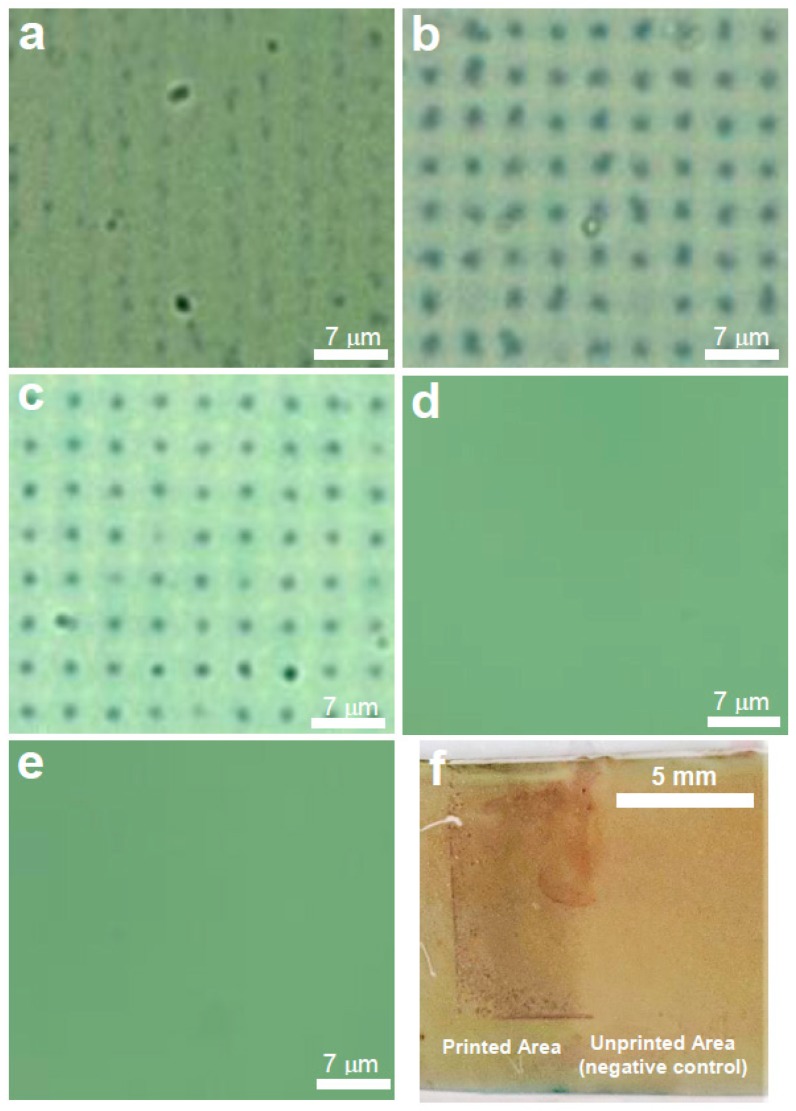
Representative optical microscopy images (40 × magnification) for immobilisation experiments on arrays with capture DNA-AuNPs and analyte. In (**a**), 10 pmol of target ssDNA was hybridised with the DNA-AuNPs at room temperature (RT) overnight and applied to the DNA array for 3 h at RT. In (**b**), 10 pmol of target ssDNA was hybridised with the DNA-AuNPs at RT overnight and applied to the DNA array for 3 h at 30 °C. In (**c**), 5 pmol of target ssDNA was tested using the same conditions as (**b**). In (**d**,**e**), only DNA-AuNPs or only 10 pmol target ssDNA was added, respectively, using the same conditions as (**b**). Figure (**f**) shows a photographic image of the slide post hybridisation using the conditions in (**b**).

**Figure 4 polymers-11-00561-f004:**
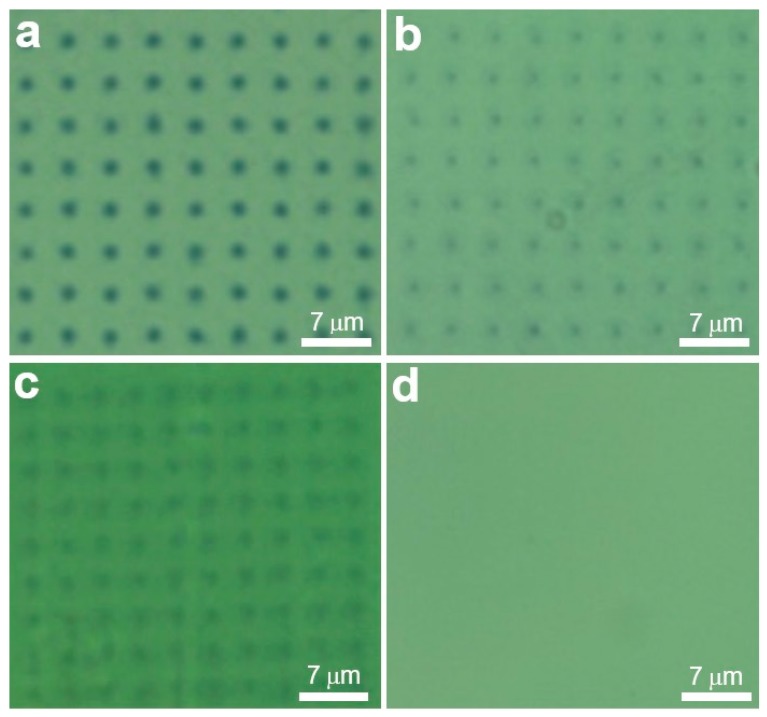
Representative optical microscopy images (40 × magnification) for immobilisation experiments on arrays with capture DNA-AuNPs and genomic DNA extracted from *G. boninense* mycelia, using (**a**) 400 ng (**b**) 160 ng, and (**c**) 30 ng and (**d**) 225 ng of non-complementary DNA extracted from soya bean. Genomic DNA was mixed with the DNA-AuNPs at 95 °C for 5 min followed by 1 h cooling at RT to allow hybridisation. The mixture was then applied to arrays for 3 h at 30 °C followed by 10 min cooling at RT prior to observation.

**Table 1 polymers-11-00561-t001:** List of ssDNA sequences used in these experiments.

Name	Sequence (5′ → 3′)
Capture probe	(Thiol C6)CCTGCTGCGTTCTTCTTCAT
Reporter probe	CGATGCGAGAGCCAA(Thiol C3)
Target	TTGGCTCTCGCATCGATGAAGAAGAACGCAGCAGG

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
