# Peer review of "Polymer Pen Lithography-Fabricated DNA Arrays for Highly Sensitive and Selective Detection of Unamplified Ganoderma Boninense DNA"

_polymers, 2019, doi:10.3390/polym11030561_

Round 1
Reviewer 1 Report
The motivation and relevance of the present work, that is to say, the use of a lithographic method that is:
a) soft to treat biological samples,
b) cost-effective,
c) yields high-throughput and
d) it is yet being able to pattern nanoscale features
to fabricate an assay to detect a pathogen that is harmful to the agricultural sector, is well pointed out. And specifically, for the Scanning Probe Lithography/ Polymer Pen Lithography field, it is also relevant to have more works where the technique is developed for the advance in bio-applications. For the above-mentioned reasons, I consider this work in principle publishable in ‘Polymers’ journal. However, I would like to express some points below that are needed to improve the manuscript, with respect to the current form:
1. Through the ‘2. Experimental section’, the authors make use of expressions such as ‘reported previously’ several times to explain some procedures or materials relevant to this work. In principle I understand why they have applied this approach. However, in the point ‘2.2. Preparation of capture DNA-AuNP conjugates’, line 117, I wonder if the authors could put an extra-effort to explain the preparation with more detail, even if has been done previously. I think this part is important for the developing of this work and therefore, it would be worthy to help the readers further, without making them to search for the corresponding reference.
2. In section ‘3.1. PPL fabrication of reporter DNA probe arrays’, I missed a figure where the scheme of the process to produce the DNA arrays is shown, such as figures 7 and 8 from the article: ‘Strategies for patterning biomolecules with dip-pen nanolithography’, Chien-Ching Wu, David N. Reinhoudt, Cees Otto, Vinod Subramaniam, and Aldrik H. Velders. Small, 7, 989-1002 (2011)
I think it would be nice if they add this reference in their list of References as well, due to the relevance of this review with the present work
3. As a reader, I find a little bit difficult to follow the panels in figures 3 and 4 with their corresponding explanations through the main text. I would suggest the author to order them in an easier way to help the readers. For example, in figure 3, putting together to the caption letters: a), b), c)…The conditions such as the temperatures and concentrations

Author Response
Reviewer #1: This reviewer’s comments were very positive. Reviewer has suggested minor
changes in the manuscript, which are mentioned below and incorporated in the revised
manuscript.
1. Through the ‘2. Experimental section’, the authors make use of expressions such as ‘reported
previously’ several times to explain some procedures or materials relevant to this work. In principle I
understand why they have applied this approach. However, in the point ‘2.2. Preparation of capture
DNA-AuNP conjugates’, line 117, I wonder if the authors could put an extra-effort to explain the
preparation with more detail, even if has been done previously. I think this part is important for the
developing of this work and therefore, it would be worthy to help the readers further, without making
them to search for the corresponding reference.
As requested by the reviewer, we have added further details in section 2.2 that now give the complete experimental procedure for the preparation of the DNA-AuNP conjugates.
2. In section ‘3.1. PPL fabrication of reporter DNA probe arrays’, I missed a figure where the scheme of the process to produce the DNA arrays is shown, such as figures 7 and 8 from the article: ‘Strategies for patterning biomolecules with dip-pen nanolithography’, Chien-Ching Wu, David N. Reinhoudt, Cees Otto, Vinod Subramaniam, and Aldrik H. Velders. Small, 7, 989-1002 (2011). I think it would be nice if they add this reference in their list of References as well, due to the relevance of this review with the present work.
As suggested by the reviewer, we have added this reference in the revised manuscript in the
introduction section as ref 8. Also, we have included the schematic of PPL patterning for the array
formation of DNA in Figure 1.
3. As a reader, I find a little bit difficult to follow the panels in figures 3 and 4 with their corresponding explanations through the main text. I would suggest the author to order them in an easier way to help the readers. For example, in figure 3, putting together to the caption letters: a), b), c)...The conditions such as the temperatures and concentrations.
As suggested by the reviewer, we have revised the captions of Figures 3 and 4 so they now include a description of the experimental conditions.
Reviewer 2 Report
This work reports a new lithography method for using polymer pen lithography to fabricate DNA oligonucleotide arrays, as well as describe the application of the arrays for the sensitive and selective detection of G. boninense, a fungal pathogen of the oil palm. The experiment and result description are clear in this report. It may make a contribution moving toward the direction of enable the fabrication of diagnostic devices for the biomedical and agri-food sectors. In my opinion, this paper can be accepted in the journal of Polymers. The small point that the authors need to attention is about the figure 2. Why state “scale bar is 25 cm in Figure 2c” in the figure legend, not show in the figure directly, like Figure 2A?
Author Response
Reviewer #2: This reviewer’s comments were very positive overall, noting that the manuscript is significant in its field.This work reports a new lithography method for using polymer pen lithography to fabricate DNA oligonucleotide arrays, as well as describe the application of the arrays for the sensitive and selective detection of G. boninense, a fungal pathogen of the oil palm. The experiment and result description are clear in this report. It may contribute moving toward the direction of enable the fabrication of diagnostic devices for the biomedical and agri-food sectors. In my opinion, this paper can be accepted in the journal of Polymers. The small point that the authors need to attention is about the figure 2. Why state “scale bar is 25 cm in Figure 2c” in the figure legend, not show in the figure directly, like Figure 2A?
We have revised the scale bar in Figure 2c in the revised manuscript.
Reviewer 3 Report
The manuscript reports the fabrication of DNA array and its application for the detection of G. boninense via polymer pen lithography. This reviewer will suggest publishing it in Polymers if the following concerns and suggestions can be fully addressed. However, the revised manuscript may need further review.
1. The authors are utilizing an unfriendly manner to introduce the concept of polymer pen lithography(PPL). They attempt to put forward the drawbacks of other approaches, and make a comparison with the advantage of PPL. Actually, as one of lithography approaches, PPL has its pros and cons.
2. The authors claim that other detection methods possess practical disadvantages in regards to complexity, cost and the need for specialist training (line 79-80). However, the method reported in this manuscript does not address these disadvantages also. The current approach is neither facile nor without needing specialist training.
3. The experimental results from lnk 1 exhibiting the lack of observable feature should be incorporated in supplementary information.
4. Regarding to annealing process for hybridization (line 245), a gradual cooling procedure is generally applied instead of heating at a constant temperature. I would suggest the authors using a cooling process like from 50 °C to room temperature, and compare with that annealed at 30 °C for 3h.
5. It is important to use atomic force microscopy (AFM) to characterize at least one pattern. AFM will provide higher resolution and more detailed information in term of NP organization within the pattern.
Author Response
Reviewer #3: This reviewer’s comments were also positive. They have however raised a number of items which we are pleased to be able to address:1. The authors are utilizing an unfriendly manner to introduce the concept of polymer pen lithography (PPL). They attempt to put forward the drawbacks of other approaches, and make a comparison with the advantage of PPL. Actually, as one of lithography approaches, PPL has its pros and cons.
It is of course true that all methods have their own pros and cons. However, this manuscript is not a critical review of the various lithographic methods, but a report on the use of PPL to produce a diagnostic assay. In case of polymer pen lithography (PPL), a scanning probe instrument and a trained operator is required to perform the lithography. Nevertheless, PPL has advantages over other lithographic techniques such electron beam lithography, micro-contact printing and dip pen lithography; as mentioned in the introductory section.
2. The authors claim that other detection methods possess practical disadvantages in regards to complexity, cost and the need for specialist training (line 79-80). However, the method reported in this manuscript does not address these disadvantages also. The current approach is neither facile nor without needing specialist training.
We do not agree with reviewer on this point. Although, one needs specialist equipment and training to use PPL and thus to prepare DNA arrays, once these arrays are produced the detection assay only involves simple operations such as mixing of the conjugates with genomic DNA, mild heating step and washing of the substrates. This assay step itself is vastly more convenient that existing methods that require PCR. We have now included a comment on this point in the conclusions section of the manuscript: “Although the fabrication of the DNA arrays involves the use of PPL that does require specialist equipment and training, once the arrays are produced these are stable, portable and easy to use in the field.”
3. The experimental results from lnk 1 exhibiting the lack of observable feature should be incorporated in supplementary information.
As requested by the reviewer, we have added the figure in the revised supplementary document.
4. Regarding to annealing process for hybridization (line 245), a gradual cooling procedure is generally applied instead of heating at a constant temperature. I would suggest the authors using a cooling process like from 50 °C to room temperature, and compare with that annealed at 30 °C for 3h.
The reviewer’s suggestion is a scientifically valid one, however gradual cooling in the manner proposed would require a specialist item of equipment and would add cost and inconvenience. This would be undesirable for an assay intended for in-field use. Our aim was to develop a procedure with maximum simplicity as far as possible.
5. It is important to use atomic force microscopy (AFM) to characterize at least one pattern. AFM will provide higher resolution and more detailed information in term of NP organization within the pattern.
This manuscript reports the development of a convenient, sensitive and selective assay for G. boninense. It is not intended to study the fine structure and organisation of the nanoparticles in the features. Indeed, such studies have already been well established in the literature for over a decade and is not novel (for example: RSC Adv. 2018, 8, 26571; ACS Nano 2010, 4, 6153; Nano Lett. 2003, 3, 1391). Furthermore, it would be logistically very difficult to carry out the proposed study since the assay development was carried out in Malaysia, where access to high-resolution AFM imaging is not available; and export of genetic material from a pathogenic organism to the UK requires government authorisation due to the biosecurity implications.
Round 2
Reviewer 3 Report
The reviewer's concerns have been addressed, and this reviewer believes this work now is good for publishing in Polymers.